# Impacts of Firm Performance on Corporate Social Responsibility Practices: The Mediation Role of Corporate Governance in Ethiopia Corporate Business

**Ma Ying, Gashaw Awoke Tikuye**  **and He Shan** *

School of Management, Wuhan University of Technology, Wuhan 430070, China; mying331@163.com (M.Y.); awoke.gashaw@yahoo.com (G.A.T.)
* Correspondence: whutheshan@163.com

**Abstract:** In today's globalized economy, the corporate company faces ever-increasing competitive and social pressures. This paper aims to identify the impacts of firms' performance on corporate social responsibility practices using the mediating roles of corporate governance evidence from Ethiopia's corporate business. The impacts of firms' performance on CSR and corporate governance as a mediator variable were studied using a sample of TIRET corporate companies, in the Amhara region, Ethiopia. The structural equation model and multiple regression analysis were estimated and tested using 21 corporate companies. The derived model reveals how corporate governance mediates the favorable relationship between CSR and firm performance. The result indicates that a firm's performance is the most significant influencing factor on CSR among the impacts examined in this study. Corporate governance has a positive role in serving as a legitimacy source for CSR practice. This study discusses the significance of results-based resource theory and presents the conclusion and implications. To solve the gaps in firm performance, return on asset, debts on capital structure, and governance, the corporate firms should identify unproductive enterprises and outsource non-core values. To overcome the existed inefficiency difficulties, this study proposed that corporate enterprises should be restructured, rebranded, reconsider their business models, and acquire technology-based firms. This paper contributes to CSR literature in the context of emerging economies. Firms, policymakers, and practitioners may take steps to improve CSR practice. In general, we conclude that in Ethiopia, including in the Amhara region, socially responsible corporate enterprises are more likely to be successful, and vice versa.

**Keywords:** corporate social responsibility; path modeling; corporate governance; TIRET corporate

## 1. Introduction

Surviving in a highly competitive market economy necessarily requires that companies focus on essential factors such as performance, governance, and social responsibility pressures for ever-increasing competitive advantages. Dynamic capabilities are the ability of a firm to combine, develop, and reconfigure external and internal expertise in order to respond to speedily changing environments [1]. Firms in similar industries perform differently because of various types of resources and capabilities [2], where the resource-based view of the organization looks at a firm's unique, rare, and imitable resources that have created a competitive advantage and expanded growth [3]. Furthermore, the relationship between dynamic capabilities and organizational performance is positive [4].

Firms are deploying a variety of significant equipment and reconfiguring business models to meet anticipated demands. The competitive advantages may be derived from firm-level resources and difficult-to-imitate corporate social responsibility actions. Corporate social responsibility (CSR) has emerged as a sustainable corporate strategy over the last few decades, whether through governmental regulation, consumer demand, or market conditions which continue to play an essential role in the global economic downturn [5].

CSR is receiving increasing attention from companies themselves, but also increasing attention from society as a whole [6].

CSR is the practice of businesses incorporating social, economic, and environmental issues into their business operations and interactions with stakeholders [7]. CSR is also defined as companies' concerns over their legal obligations towards society and environmental effects, which focus on sustainable development, public and environmental policy [8]. CSR is broadly defined as an approach to diminishing the negative consequences of corporate production, operation and ensuring society's welfare and compassionate practices of businesses under pressure by owners and shareholders to improve profitability rather than build or preserve organizations [9]. Furthermore, CSR is a concept in which firms integrate social and environmental concerns into their business operations and interact with their voluntary and mandatory activities [10]. It is ethics, citizenship, good corporate governance, and others [11]. As the above literature shows, CSR has various concepts and contextual definitions. Therefore, to the extent that the CSR knowledge base is limited in terms of understanding, availability, and tacitness, CSR's successful adoption commonly depends upon firms and countries developing capabilities. All these confirm that issues concerning CSR concepts are still being debated.

Various researchers have explored the links between CSR and firm performance using diversified approaches. Some studies show mixed (positive, negative, or neutral) results regarding CSR and firm performance. CSR positively affects corporate performance for firms engaged in CSR, whereas firms' social irresponsibility activities reduce their performance [12]. CSR positively and significantly influences a firms' performance indicators, specifically return on assets (ROA), returns on equity (ROE), and earnings per share ratios [13]. Simultaneously, CSR dimensions (environment, customers, suppliers, employees, and social) relate positively to firms' performance. CSR has a significant and positive association with firm performance [14], whereas the mediating effect of Chief Executive Officer (CEO) and ownership positively correlates to firms' performance and ownership with CSR. Furthermore, CSR moderates significantly and positively between corporate governance and firms' performance [15]. The firms fulfilling CSR would significantly impact firm performance [16]. According to the previous findings, there is a significant relationship between companies' performance and their CSR [17]. The CSR not only boosts the company's social value and reputation, but it also improves profitability and performance. Firm performance has a statistically significant impact on CSR, and companies with better financial performance also undertake more CSR practices [18].

In contrast, CSR and financial performance may have a negative relationship. Firms disclose more information on CSR initiatives when they have lower returns on assets [19]; after controlling for the firms' debt and size, highly levered firms are less profitable, and larger firms have higher profits. That is, there is no substantial relationship between CSR and corporate performance [20]. The causal relationships between CSR and financial performance reveal that greater social responsibility does not lead to better financial performance, and financial performance has a negative impact on corporate social responsibility [21]. The possible vice versa influence of company performance on CSR was found to have a mixed connection [22]. These findings revealed a positive association between operational dimensions of corporate social responsibility and firm performance, as well as a negative relationship between non-operational elements of corporate social responsibility and firm performance.

Hence, previous studies have yielded inconsistent results. While some outcomes appear plausible, others contradict one another and lead to different conclusions. Some of these studies used panel data, while others used survey data. There is a methodological gap in the use of research approach, sample, instruments, and models. Furthermore, most studies did not consider corporate governance as a mediating variable in depth and they did not mainly focus on the impact of firm performance towards CSR at the corporate or enterprise level.

Some literature analyzes the influence of firm performance, firms' CSR practice, and CSR activities on firms' value using descriptive modes from the selected study area context. The causal relationship between stakeholders and CSR execution has been examined [23]. These findings reveal that the environment, customer, shareholder, and community significantly affect CSR. Evidence of CSR practices, determinants, and challenges from theoretical and empirical lessons have been reviewed [24]. This study concluded that CSR centers should attempt to foster CSR and promote academic study, encouraging the private sector within the framework of responsible business practice that creates awareness. Furthermore, the CSR practices concerning the CSR triple bottom line and which focus on the people and planet perspectives have been explored [25]. These findings conclude that there is a lack of balanced CSR practice in environmental and social CSR aspects; there should be strong community engagement and effective public relations. CSR learning in selected firms has been investigated in a qualitative case study [26]. The findings showed that firms' learning social responsibility is at the emergence stage with the state and foreign market pressure as critical motivators. While regulating environmental and labor conditions, the state offers incentives for higher economic responsibility of firms.

To summarize, the studies reviewed above show a severe limitation in focusing on the impacts of firm performance on CSR practice in the Ethiopia context. Because of this, there is room for more research into the relationship between firm performance, CSR practices, and corporate governance issues. No one has focused on how a firm's performance affects CSR, taking into account corporate governance's mediation role. Moreover, there are methodological gaps; most studies concentrate on descriptive methods rather than empirical approaches.

This paper proposes a framework to identify the impacts of firm performance on CSR using corporate governance's mediation role. Furthermore, this study used a path model analysis combining a structural equation model (SEM). In the study area context, CSR's problem lies at the company, the public, and the government level for which both are less aware of their roles, rights, and responsibilities. Ethiopia is far behind developed countries in terms of industrialization, firm performance, CSR awareness, and corporate governance. However, there is still a need to build awareness of CSR's benefits for enterprises, corporate business companies, and stakeholders. Moreover, CSR has not yet been sufficiently implemented and studied in Amhara Region, Ethiopia; for this reason, firms are taking CSR as a liability instead of a source of long-term benefits for firms, public, and environment. Therefore, there is an intense interest in studying this critical issue and understanding enterprises' socially responsible behavior in the selected study area. The impacts of firms' performance on CSR adoption using corporate governance have still not been well exploited, and CSR is not successful. The difficulty of the corporate companies in CSR activities mainly relates to firms' performance and corporate governance gaps. Thus, the deep-rooted problem becomes complicated, and the corporate companies have been exposed to critical challenges on their competitive advantage and CSR execution. These difficulties initiated and inspired the researcher to research this area. Hence, the study's main objective is to identify the impacts of firms' performance on CSR practice using the mediation role of corporate governance evidence from the Amhara Region corporate company, Ethiopia. The study also explores the impacts of firms' performance, CSR, and corporate governance accordingly. Furthermore, this study tried to identify and address the following three primary research questions based on the stated objectives: (1) What is the influence of firm performance on corporate social responsibility practices? (2) How does the mediation role of corporate governance influence firms' performance and CSR practice? (3) Is there a relationship between the impacts of firm performance, CSR, and corporate governance?

## 2. Literature Review and Hypothesis Development

### 2.1. The Evolution and Development of CSR

As various literature shows, CSR was initially proposed by Sheldon in 1924 in that protecting society's interests is an enterprise's primary responsibility when following the profit motive [27]. Furthermore, the labor conflicts that emerged at the end of the 19th century due to the industrial revolution, when the paradigm of artisan work became one of mass production, exposed a series of social problems that pushed companies to take actions that could be considered the root of CSR [28].

The capitalist model, which promoted profit maximization and self-regulation of markets during the 1950s and 1960s, exposed negligent behavior that resulted in companies' violations of human and labor rights. In the context of this fact, voices have emerged in society demanding more responsible business operations concerning social aspects [29]. Furthermore, the 1970s were also affected by a severe economic crisis, leading to the rise of social movements that played a fundamental role in bringing environmental and civil rights issues to businesses and companies [30].

In the 1980s and 1990s, concern began to be raised about the effect of human actions, including the environment, human and labor issues, and various summits of international organizations. In these moments, companies' CSR policies were used to communicate their social and environmental policies, practice and performance as well as to enhance companies' image, prestige, and social legitimization to act [31].

The globalization of markets and the freedom to operate on the side of companies in the 2000s and the increased complexity of corporate relations with different social groups or interest groups contributed to a further shift towards reforming the company model [32]. In general, CSR became a fundamental element in companies' responses to various social requirements [33], understood as to how companies assumed social commitments and responsibilities, taking into account the impact of their operations on stakeholders. CSR's movement in the new millennium indicates the concern for sustainability. CSR has been treated as an enterprise's commitment to maximizing long-term positive effects and minimizing society's negative impacts.

From the development perspective, the triple bottom line emphasizes three issues, i.e., social responsibility (people), environmental responsibility (planet), and economic responsibility (profit), from the development perspective [26]. Therefore, the TBL believes that companies should concentrate on social and environmental issues as much as they do on profits. Furthermore, a socially responsible company can be considered as an institution for economic prosperity, social equity, and environmental protection [34]. Similarly, the three CSR areas recurring in most definitions are economic, social, and ecological [35,36]. Therefore, in terms of growth, several scholars have concluded that the three dimensions are interconnected and that joint action ensures CSR's long-term sustainability.

### 2.2. The Effects of Corporate Social Responsibility

The concept of corporate social responsibility has different meanings; the debate over CSR goes back to the 1950s and there are no widely agreed meanings [37]. Various businesses, scholars, and organizations focus on CSR in multiple aspects [38]. CSR is defined as the actions that appear to further some social good beyond the firm's interests and are required by law [39]. CSR is concerned with the relationship between companies and their stakeholders [40]. Furthermore, CSR is the relationship of organizations with society and organizations' need to align their values with societal expectations [41]. Businesses should look outside of their core economic and legal responsibilities [29]. CSR is broad and grounded in understanding the company being part of society [42]. It is defined as the voluntary integration of social and environmental concerns in business operations and their interaction with stakeholders [43]. Recently, CSR has been redefined as enterprises' responsibility for their impacts on society [44].

As Brinkmann and Peattie [45] stated, CSR is a process to integrate social, environmental, ethical, human rights, and consumer concerns into the business operations and

core strategy in close cooperation with the enterprises' multi-stakeholders. CSR could be corporate conscience, social success, corporate citizenship, or sustainable and responsible business. CSR determines the companies to maximize their positive effect on stakeholders while mitigating its negative social impact [46]. Socially, companies should not be limited to philanthropic expenses turned into investment in sustainable contribution to society, and should carry out their economic operations at the same time and, as a consequence, the integration of CSR leads to profit maximization allocated to the value creation [47,48].

As stakeholder theories indicate, CSR has been created to shift the corporate focus towards mitigating undesirable operation consequences and improving social wellbeing [49,50]. Companies can exaggerate CSR initiatives to deceive customers and create credibility and confidence by misleading the environment to maximize profitability rather than improving society [51]. CSR positively affects corporate performance, whereas firms' social irresponsibility activities reduce firms' performance [12]. Moreover, all the CSR economic, social, and environmental dimensions relate positively to firm performance [13]. CSR from resource-based perspectives and CSR initiative implementation can lead to decreased operating costs and increased revenue from grants and incentives [52]. For instance, companies that adopt environmental initiatives to reduce waste, reuse materials, recycle, and conserve water and electricity can frequently obtain grants and incentives. On the other hand, CSR engagement fosters management competencies (problem solving, discovering sources of inefficiency and incentives); social responsibility management competencies might also lead to better management.

To generalize, a socially responsible company serves these needs of society, increases its goodwill, and provides a long-term and sustainable demand for its products [53]. Furthermore, CSR has long been a way of importing and integrating the effect on the environment, economy, society, and all stakeholders into company activities. Hence, CSR's significant effects are the 'triple bottom line', which takes care of people, the planet, and profit. That is why, today, many companies have rebranded their core values to include CSR. This study intends to examine the impact of firms' performance on CSR using the mediating role of corporate governance. Therefore, this study proposes the stated hypotheses concerning the relationship between CSR and firms' performance.

*2.3. The Relationship of Firms' Performance, Corporate Governance, and CSR*

2.3.1. Firms' Performance and CSR

Various studies have analyzed CSR and firm performance using different approaches to examine the impacts and relationships between firm performance and CSR. In this study, the conceptual framework combined the effects of firm performance on CSR and corporate governance mediation. CSR's dimensions positively and significantly influence the company's vision [13]. CSR performance has a positive relationship with firm performance when the performance indicators benefit growth, total asset, corporate soundness, and social contribution, which will increase the use of better CSR [54].

Companies use CSR and green business practices to encourage creativity to inspire and enhance corporate social performance [55]. A good relationship between company performance and CSR proves that companies' direct costs are not hidden fees for stakeholders. It shows that firm performance and CSR has a positive connection. In contrast, if the interest of the stakeholders and their social expectations (environment, consumers, and employees) is taken into account, the expense of CSR activities used by the company will be much lower than the CSR benefits [56]. Thus, if businesses consider CSR seriously, competitiveness raises the costs and reduces the prices concealed from the stakeholders.

The profitability of corporate financial performance has been measured by using three ROA, ROE, and ROS scales [57]. For that, there is a positive relationship and mutual reinforcement between financial performance and social responsibility. On the other hand, the causal relationships among financial performance and social responsibility have been examined [58]. As a result, more generous social responsibility does not result in better financial performance, and financial performance negatively impacts CSR. Similarly, a

meta-analytical investigation on the relationship between corporate social and economic performance reveals that corporate social performance positively impacts corporate financial performance [59].

Moreover, the relationship between CSR and a firm's performance has been explored using accounting-based measures, including return on assets (ROA), total assets, and sales growth, concludes a positive relationship between CSR and firms' performance [60]. Returns on asset (ROA), return on equity (ROE), and return on sales (ROS) affect firms' performance [61]. Better corporate social understanding leads to improved corporate financial performance [62]. In contrast, Selcuk and Kiymaz [19] explored a negative relationship between CSR and financial results in which companies that disclose more information about CSR initiatives have a lower return on assets. The study findings suggested that larger companies have higher profits after adjusting for their debt and size, whereas highly leveraged firms are less profitable. The capital structure on firms' performance view, the short-term debt, the long-term debt, and the company leverage (LEV) negatively affected assets' return [63]. Return on equity (ROE) has a negative relationship with the capital structure variables, but it is insignificant compared with the long-term debt and the company leverage. The level of liability on the capital structure negatively affects the performance of the company. To summarize, this study attempted to examine and identify the impacts of firm performance on CSR, considering all dimensions of the outcome and predictor variable indicators. Therefore, the study proposed the hypothesis as follows.

**Hypothesis 1 (H1).** *Firm performance has positive and substantial impacts on CSR practice.*

### 2.3.2. Corporate Governance and Firm Performance

The impact of corporate governance is expected to affect the firms' performance, which is counted as one of the primary issues for stakeholders since it allows them to identify the factors that influence performance and use those aspects as indicators for a firm's success or failure. In this regard, Fallatah and Dickins [64] investigates the relationship between corporate governance and firm's performance, concluding that corporate governance considerably improves firm's performance. On the contrary, Ahmed and Hamdan [65] explores the relationship between corporate governance and firm performance and concludes that the two are unconnected. However, Alsurayyi and Alsughayer [66] examines the impact of corporate governance on firm performance and determines that corporate governance is strongly linked to firm performance. Furthermore, Del Miras-Rodríguez and Martínez-Martínez [67] looked at the impact of good corporate governance on the performance of publicly listed companies, and found that proper corporate governance has a positive impact on firm performance. Both effective corporate governance and CSR have a positive effect on financial performance as well as CSR on financial performance [68,69] found that institutional regulation has a positive and significant effect on corporate governance and firm performance. The increase in corporate governance best practices influences company performance. Therefore, according to the above findings, there is still a debate on the relationship between corporate governance and firm performance. As a result, this research looked into the effects of corporate governance on firms' performance while considering its mediating role.

**Hypothesis 2 (H2).** *Corporate governance positively and significantly influences firm performance.*

### 2.3.3. The Mediation Role of Corporate Governance on Firm Performance and CSR

CSR has a significant positive relationship with a firm's performance. The relationship between CSR and firms' performance shows the same results as board interaction [14]. The interaction between management ownership and CSR has a significant positive relationship with the firm's performance while the interaction between the concentration of ownership and CSR has a positive effect on the firm's performance. Furthermore, cor-

porate governance practices are positively and significantly associated with the level of CSR initiatives [70]. It enables the organization and statutory bodies to consider corporate governance practices, which will enhance CSR initiatives.

The main factors determining CSR engagement's strength at the firm level are the structure of equity ownership, the board of directors' composition, and the regulatory framework on corporate governance and CSR [71]. Larger firms tend to have more resources than small and medium firms in terms of capital and talent; hence, they can make considerable investments in CSR activities [72]. Corporate governance positively and significantly affects CSR. In this case, firms' efficient corporate governance mechanisms help improve associated firms' corporate social responsibility practices. As a mediator effect, the mechanism of good corporate governance and CSR has a positive effect on financial performance, CSR, and firm performance [68]. Moreover, the government has a determining role in motivating and influencing CSR practices [73]. Hence, this study tried to examine corporate governance as a mediator role to explore the impacts of firm performance on CSR evidence from Ethiopia corporate business, Amhara region TIRET corporate state-owned endowment enterprises. Therefore, the study argues that corporate governance positively and significantly affects the firm's performance and CSR as a mediation role.

**Hypothesis 3 (H3).** *Corporate governance positively and significantly mediates firm performance and CSR.*

### 2.4. Conceptual Model of the Study

A conceptual model is a blueprint or guide for research to help the researchers organize, conceptualize, and conduct their research, whether qualitative, quantitative, or mixed methods [74,75]. This study proposed a research model which was designed to investigate the impacts of firms' performance on CSR by using corporate governance as a mediator variable empirically. This conceptual model includes the independent variable firm performance, the mediator variable corporate governance, and the dependent variable CSR practice. With these assumptions, return on asset (ROA), return on equity (ROE), return on sales (ROS), and liability on capital structure are indicators of firm performance. Similarly, board, ownership, audit, and transparence are indicators of corporate governance. The economic, social, and environmental dimensions are indicators of the outcome variable CSR practice. Therefore, all study variables have been represented in the conceptual model, as shown in Figure 1.

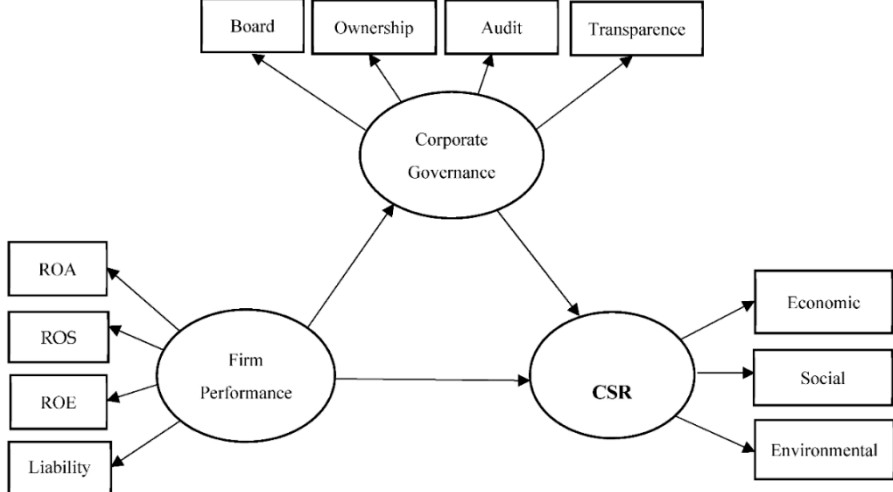

**Figure 1.** Authors' proposed conceptual model for the study.

## 3. Research Methodology

### 3.1. Study Design

In this study, a quantitative and qualitative research approach was applied [76]. The nature of the research aim and question determines the researcher's rationality to choose a specific research design, the data to be used, the data to be collected, analytical tools and skill to be deployed, the existing resource at hand, and other related considerations. The proposed research model in the conceptual framework is adapted from previously examined studies [10,64,77]. Because of the mediator, variable and measurement instrument, model, and methodological differences, this study differs from the adapted research design. The significantly improved model includes corporate governance as a mediator variable with indicators such as board, audit, and transparency. In contrast, liability on capital structure was provided as an indicator for firm performance. This study used both primary and secondary sources of data. The primary data sources were obtained from the selected respondents, such as corporate boards, management, and staff members. The secondary data were collected from reports, surveys, policy documents, and literature. The study considered the impacts of firm performance as the predictor variable, CSR as an outcome variable, and the role of corporate governance as a mediation variable. In general, the study used 42 standardized structured questionnaires and a Likert five-point scale (1 = strongly disagree, 2 = disagree, 3 = neutral, 4 = agree, 5 = strongly agree) was used to measure the questionnaire items. This study used data from corporate companies from the seven years from 2008 to 2015 to evaluate firms' performance [78].

### 3.2. Sample and Sampling Techniques

In the Amhara Region, the TIRET corporate company has 17 state-owned endowment enterprises and 4 enterprises shared with private owners. The researcher took the sample size of all 21 corporate business enterprises. This study tried to control the possibility of bias and sampling errors. The researchers used individual respondents from the population of 6060 employees to collect information, and the determined sample size is 357. The sample utilized using non-probability purposive sampling techniques to categorize the area and estimate the respondents' number [79]. The number of clusters from the total number of companies was determined using targeted sampling techniques. The study also used simple random sampling techniques to choose respondents. By selecting individual participants to fill in the questionnaires, the research also applied simple random sampling techniques. A sample size of 357 was decided based on the Morgan formula determination of estimating sample size [80].

### 3.3. Methods of Data Analysis

The collected data were coded, inserted, and processed using SPSS/AMOS statistical software for all the required analytical techniques. Nominal scales were used to measure all instruments. The study also used the marketing-measurement approach to evaluate firm performance results. The derived hypotheses were tested using chi-square goodness-of-fit tests. The study used AMOS to estimate all latent variable coefficients on the structural equation model, using path diagram analysis. The research also used trend analysis techniques, depending on the corporate company's growth path report from 2008–2015.

### 3.4. The Structural Equation Model (SEM) Specification

The structural equation (SEM) analysis is designed to test a single analysis model instead of trying separate regression analyses. SEM is an important technique to test the direct and mediating effect on the direct and indirect impact [81]. All variables were hypothesized for causal estimation in which one variable affects a second variable that affects a third variable. Therefore, the intervening variable, $M$, is the mediator. It mediates the relationship between predictor $X$ (firm performance) and $Y$, an outcome variable. The derived equation was formulated from this view as follows.

First, to estimate the impacts of firm performance on CSR, the dependent variable "$Y$" was predicted as:

$$Y = \beta_0 + \beta_1 X + \varepsilon \tag{1}$$

Second, the impacts of firm performance on CSR using corporate governance's mediation role among them were predicted through multiple regression analysis with the "$X$" and "$M$" predicting equation.

$$Y = \beta_0 + \beta_1 X + \beta_2 M + \varepsilon \tag{2}$$

Third, the impact of mediator variable corporate governance between firm performance and CSR was predicted as follows:

$$Y = \beta_0 + \beta X + \varepsilon \tag{3}$$

where: $Y$ = the dependent variable (CSR), $X$ = the independent variable (Firm Performance), $M$ = the mediator variable (Corporate Governance), $\beta_0$ = Intercept, $\varepsilon$ = the standard error.

*3.5. Measurements of Variables*

3.5.1. Dependent Variables

The measurements of questionnaires consist of three parts adopted in the context of study variables. The dependent variable CSR is measured by economic, social, and environmental dimension instruments [82–86]. The measurement instrument indicators were adopted from the context of the study. These are nine economic indicators (stakeholder involvement, the response of customers compliance, quality of products, customer satisfaction, maximizing profit, minimizing operating costs, monitoring employee productivity, engaging in long-term business), six social dimensions (training and education, human rights, community development, health, safety in the workplace, employment and labor relation) and five environmental dimensions (pollution, energy, waste, transport, ecological compliance). CSR was measured using indicators estimated through a questionnaire containing information about the economic, environmental, and social dimensions.

3.5.2. Predictor and Mediator Variables

In this study, researchers used economic and financial indicator variables to measure firms' performance. The study also considered such measures as objective and consistent with the sample, with the information analyzed as comprising the corporate company state-owned endowments and enterprises. With this in mind, the marketing approach performance indicators measure the independent variable of firm performance [87–89]. Hence, the firms' performance indicators were adapted from the objective of the study aligned to ROA, ROE, ROS, and liability on the capital structure or debt ratio. Therefore, this work used total sales, total assets, total liability, total capital, total equity, and financial strategic year results. These values allow, in order, measuring of the levels of internal performance of the corporate company at the enterprises level with respect to the social expectations. The mediating variable indicator questionnaires were adopted and used to evaluate the role of corporate governance on firm performance and CSR practices [10,90,91]. Therefore, the measurement items for the mediator variable corporate governance are board, audit, ownership, and transparency. To sum up, all the determined variables were measured depending on the collected primary and secondary data after all the analyses of estimated results, validating, verifying and crosschecking their expected requirements.

## 4. Results and Interpretation

*4.1. Analysis of Demographic Data*

In this study, the findings of 357 sample respondents' demographic data indicate that 20.2% (72) and 79.8% (285) of respondents were female and male, respectively. Here, the statistical data implies that female participation is too low; this may be related to the

corporate business firms' inability to give more attention to female employees. The age of respondent analysis shows that 36.7% of the respondents were 31–40 years old, 25.8% were 41–50 years old, 21.3% of respondents were between 21–30 years old, and 16.2% of the respondents were more than 50 years old. It shows that two-thirds of the respondents were mature and productive. The respondents were able to understand and give a reasonable response to the researcher. In the respondents' educational background, most respondents, 49.6%, were undergraduates (degree holders), 19.9% of the respondents had diplomas, 14.6% of respondents were postgraduates, and the remaining 16% of respondents were at the high school level. This implies most of the respondents were able to give valuable information to the study.

### 4.2. Exploratory Factor Analysis, Validity and Reliability

All the necessary reliability analysis considered the data validation standards to fulfill the extent of measurement instruments for data accuracy and reliability. Hence, the prominent reliability measurement is Cronbach's alpha to measure the measurement instruments' internal consistency and reliability. The Cronbach's alpha score should be greater than or equal to 0.70 (alternatively, 0.80); alpha is the best choice among all reliability coefficients to meet the internal consistency preconditions [92]. Therefore, the coefficient values of the alpha score ranged from 0.711 to 0.885. This implies that all the calculated variables for the measurement model of the outcome variable corporate social responsibility (CSR) constructs are the best fit, valid, and reliable.

Exploratory factor analysis (EFA) is used to determine the validity of the instruments and to organize items into the constructs under one specific variable [93]. The composite reliability (CR) of the latent factors estimate of reliability ranged from 0.861 to 0.923, which indicates the scale of CR level of reliability has a reasonable internal consistency and is acceptable. The value of the critical ratio (CR) > 0.70 is accepted [94]. Similarly, the estimated value of the average variance extracted (AVE) for the constructs ranged from 0.51 to 0.80 as shown in Table 1, which is considered very good because the level of AVE value was >0.50 of the recommended values [95].

**Table 1.** Construct validity and reliability analysis.

| Construct | Indicators | Factor Loadings | Cronbach's $\alpha$ | Composite Reliability (CR) | Average Variance Extracted (AVE) |
|---|---|---|---|---|---|
| CSR | ECOND | 0.881 | 0.736 | 0.923 | 0.80 |
| | SD | 0.909 | 0.817 | | |
| | END | 0.892 | 0.809 | | |
| Firm Performance | ROA | 0.834 | 0.711 | 0.861 | 0.51 |
| | ROE | 0.769 | 0.792 | | |
| | ROS | 0.857 | 0.775 | | |
| | LIB | 0.893 | 0.783 | | |
| Corporate Governance | BD | 0.806 | 0.789 | 0.869 | 0.52 |
| | OS | 0.854 | 0.727 | | |
| | AD | 0.832 | 0.885 | | |
| | TP | 0.831 | 0.780 | | |

### 4.3. Regression Analysis

According to the results, each parameter estimate value ranges from 0.892 (Corporate Governance) to 1.665 (Firm Performance), and all are positive. When the estimates are separated by their relevant standard error (S.E), they create critical ratio values (C.R). A critical ratio (C.R) score that is larger than 1.96 is significant at the *p*-value 0.05 level. All critical ratios (C.R) values are greater than 1.96, which shows a statistically significant level at *p*-value 0.05. Each value of the variables was tested independently to verify the fitness of the model. As a result, the independent variable firm performance's significance level is higher than the *p*-value of 0.05. The *p*-value of the mediator variable corporate

governance is 0.05, and the outcome variable CSR level of significance is 0.05. This shows that there is no significant disparity in the variances of all variables. Therefore, when the alpha criteria are below 0.05 or 0.001, the *p*-value is significant. It is important if the *p*-value falls below the set threshold. Furthermore, if the estimated *p*-value goes to ***, the variables are considered to be highly significant and have a positive response to one another as shown in Table 2 with the satisfactory support of path directions.

**Table 2.** Regression weights for the level of significant and critical ratio.

| List of All Variables | | Path | Estimate | S.E. | C.R. | *p* |
|---|---|---|---|---|---|---|
| Corporate Governance | <— | Firm Performance | 1.471 | 0.215 | 6.837 | *** |
| CSR | <— | Corporate Governance | 1.488 | 0.205 | 7.243 | *** |
| CSR | <— | Firm Performance | 1.500 | 0.289 | 5.181 | *** |
| ROA | <— | Firm Performance | 1.000 | | | |
| ROE | <— | Firm Performance | 1.291 | 0.199 | 6.505 | *** |
| ROS | <— | Firm Performance | 1.665 | 0.224 | 7.449 | *** |
| LIB | <— | Firm Performance | 0.896 | 0.120 | 1.999 | 0.046 |
| BD | <— | Corporate Governance | 1.000 | | | |
| OS | <— | Corporate Governance | 0.961 | 0.068 | 14.103 | *** |
| AD | <— | Corporate Governance | 1.014 | 0.073 | 13.964 | *** |
| TP | <— | Corporate Governance | 0.979 | 0.073 | 13.459 | *** |
| ECOND | <— | CSR | 1.000 | | | |
| SD | <— | CSR | 1.197 | 0.068 | 17.671 | *** |
| END | <— | CSR | 1.189 | 0.072 | 16.592 | *** |

Note: *** *p*-value stands for strong level of significant at *** *p*-value < 0.001.

### 4.4. Analysis of Mediating Effect

Predictions are concerned with the role of corporate governance as a mediating variable. Considering the path model, AMOS was utilized to analyze the mediation effect of the study variable. To test the mediation effect in the structural equation model (SEM), the full mediation model was compared to a partial mediation model in which direct paths from the independent variables were added to the dependent variable. Hence, direct, indirect, and total effects of the mediator variable corporate governance were analyzed, including the mediation effect findings, as shown in Table 3.

**Table 3.** The direct, indirect and total effects of the mediator variable corporate governance.

| List of Variables | Corporate Governance | | | Firm Performance | | | CSR | | | Findings on Mediation Effects |
|---|---|---|---|---|---|---|---|---|---|---|
| | Direct Effects | Indirect Effects | Total Effects | Direct Effects | Indirect Effects | Total Effects | Direct Effects | Indirect Effects | Total Effects | |
| Corp. Gov. | 0.000 | 0.000 | 0.000 | 1.471 | 0.000 | 0.841 | 0.000 | 0.000 | 0.000 | Supported |
| CSR | 0.000 | 0.000 | 0.000 | 1.488 | 0.000 | 0.776 | 0.000 | 0.000 | 0.000 | Supported |
| END | 0.000 | 0.000 | 0.000 | 0.000 | 0.604 | 0.604 | 1.189 | 0.000 | 0.825 | Supported |
| SD | 0.000 | 0.000 | 0.000 | 0.000 | 0.650 | 0.650 | 1.197 | 0.000 | 0.888 | Supported |
| ECOND | 0.000 | 0.000 | 0.000 | 0.000 | 0.579 | 0.579 | 1.000 | 0.000 | 0.792 | Supported |
| TP | 0.979 | 0.000 | 0.745 | 0.000 | 0.627 | 0.627 | 0.000 | 0.000 | 0.000 | Supported |
| AD | 1.014 | 0.000 | 0.774 | 0.000 | 0.651 | 0.651 | 0.000 | 0.000 | 0.000 | Supported |
| OS | 0.961 | 0.000 | 0.781 | 0.000 | 0.657 | 0.657 | 0.000 | 0.000 | 0.000 | Supported |
| BD | 1.000 | 0.000 | 0.743 | 0.000 | 0.625 | 0.625 | 0.000 | 0.000 | 0.000 | Supported |
| LIB | 0.896 | 0.000 | 0.000 | 1.500 | 0.000 | 0.731 | 0.000 | 0.000 | 0.000 | Supported |
| ROS | 0.000 | 0.000 | 0.000 | 1.665 | 0.000 | 0.884 | 0.000 | 0.000 | 0.000 | Supported |
| ROE | 0.000 | 0.000 | 0.000 | 1.291 | 0.000 | 0.566 | 0.000 | 0.000 | 0.000 | Supported |
| ROA | 0.000 | 0.000 | 0.000 | 1.000 | 0.000 | 0.397 | 0.000 | 0.000 | 0.000 | Supported |

To conclude, the mediation effect of corporate governance between firm performance and CSR was seen from two paths. The first one is the path between firms' performance to

corporate governance. The mediating effects of corporate governance on firm performance are powerful, positive, and significant ($\beta$ = 0.861 ***). The outcome implies that the performance of firms is highly determined by the role of corporate governance. The other is the link between corporate governance and CSR. Here, CSR is positively and significantly influenced by corporate governance engagement ($\beta$ = 0.767 ***). The result indicates the intervening legitimate supporting, controlling, and monitoring practice of corporate governance on CSR is very influential next to the impacts of firm performance. The mediating effects of corporate governance between firms' performance and the outcome variable CSR are very significant at all dimensions of direct, indirect, and total impact. Moreover, the mediating effects of corporate governance have a multiplier effect on both paths. The cumulative impact of corporate governance has a dual intervening legitimacy for supporting and implementing CSR, as shown in Table 4.

**Table 4.** The mediating effects of corporate governance on firm performance and CSR.

| Mediating Variable | Path | Effects of Corporate Governance | | | |
|---|---|---|---|---|---|
| | | Sobel Test | Mediation Effect | LL95%CI | UL95%CI |
| Corporate Governance | FP –> CG–> CSR | 2.73392591 | 0.861 | 0.555 | 0.744 |
| | | | 0.767 | 0.322 | 0.592 |

*4.5. Model Estimate and Test of Hypothesis*

4.5.1. Model Fit Indices (MFI) Result Analysis

To estimate the impacts of predictor variables on the outcome variables, we used a supporting path diagram SEM. As the model summary fit the indices, all of the measurement, criteria, and the estimated values indicate that the model is appropriate and fits. The recommended criterion values (GFI, AGFI, NFI, IFI, TLI, CFI, RMSEA, and chi-square ($X^2$)) adequately fulfilled the expected requirements. The derived model performance was tested and validated based on previously recommended standards. This implies that the proposed model might be adaptable for both state-owned and non-state-owned firms as the objective and nature of the business. Therefore, the summary of the model fitness result enables the decision to accept or reject measures, as shown in Table 5.

**Table 5.** Summary of model fit indices.

| Model Fitness Indices | Recommended Values | Structured Model |
|---|---|---|
| Chi-square ($X^2$) | Low | 684.458 |
| Probability | >0.05 | 0.000 |
| Degree of freedom (df) | >0.0 | 9 |
| $X^2$/df | - | 76.05 |
| Goodness of Fit Index (GFI) | >0.90 | 0.895 |
| Adjusted GFI (AGFI) | >0.90 | 0.895 |
| Normed Fit Index (NFI) | >0.90 | 0.901 |
| Incremental Fit Index (IFI) | >0.90 | 0.918 |
| Tucker Lewis Index (TLI) | >0.90 | 0.898 |
| Comparative Fit Index (CFI) | >0.90 | 0.917 |
| Root Mean Square Error of Approximation (RMSEA) | <0.080 | 0.047 |

Note: The research model is fit.

4.5.2. Estimates of Coefficients and the Structural Path Modeling Analysis

To carry out the evaluation of the structural equation model (SEM), the study considered the values of the path coefficients or standardized regression weights ($\beta$) and the explained variance ($R^2$). To determine the derived hypotheses of the model, nonparametric resampling techniques were used to examine the stability of estimates offered by SEM

(nonparametric bootstrap technique). The study's findings result in a model that examined the impacts of firm performance on CSR using corporate governance's mediation role. The variables' standardized coefficient regression estimate values indicate the model fits, because the estimated value of R square ($R^2$) is 0.861 (86.1%) and significant at *p*-value (0.001). The path diagram model revealed that the independent variable firms' performance affects CSR ($\beta$ = 0.956 ***) and CSR is affected by the mediator variable corporate governance ($\beta$ = 0.841 ***). The inferences of all indicator measurement instrument standardized estimate coefficients ($\beta$) confirmed the extent to which each indicator influences the predictor, mediator, and outcome variables, as shown in Table 6.

**Table 6.** Standardized estimates regression for structural (path) model analysis.

| Coefficients | Path | | Estimate ($\beta$) |
|---|---|---|---|
| Corporate Governance | <— | Firm Performance | 0.841 |
| CSR | <— | Corporate Governance | 0.776 |
| CSR | <— | Firm Performance | 0.956 |
| ROA | <— | Firm Performance | 0.397 |
| ROE | <— | Firm Performance | 0.566 |
| ROS | <— | Firm Performance | 0.884 |
| LIB | <— | Firm Performance | 0.267 |
| BD | <— | Corporate Governance | 0.743 |
| OS | <— | Corporate Governance | 0.781 |
| AD | <— | Corporate Governance | 0.774 |
| TP | <— | Corporate Governance | 0.745 |
| ECOND | <— | CSR | 0.792 |
| SD | <— | CSR | 0.888 |
| END | <— | CSR | 0.825 |

Note: ROA = Return on asset, ROE = Return on equity, ROS = Return on sales, LIB = Liability on capital structure; BD = Board, OS = Ownership, AD = Audit, TP = Transparence; ECOND = Economic dimension, SD = Social dimension, END = Environmental dimension are indicators for Firm performance, Corporate governance, and CSR, respectively.

The impacts of firm performance: The impacts of firm performance on CSR depend on the estimated verified variance and covariance coefficient values in the model. The predictor variable firm performance is affected by return on sales ($\beta$ = 0.884 ***), return on equity ($\beta$ = 0.566 ***), return on asset (0.397 ***), and liability on capital structure (0.267 ***), respectively. As the remain variables are constant, when the firms' ROS, ROE, ROA, and liability on capital structure increase by one unit, the firm performance also shifts by one unit. Therefore, when the impacts of firm performance increase by one unit, the outcome variable CSR increases by the estimated factor of $\beta$ = 0.956 significantly at *p*-value (0.001).

Corporate governance's role in firms' performance: The mediator variable corporate governance affects firm performance by the estimated factor of $\beta$ = 0.841 **. We assume that everything else is constant, so when the corporate board, ownership, audit, and transparency shifts by one unit, the role of corporate governance improves by one unit. Hence, as corporate governance's role increases, the firm performance also increases by the estimated factor of $\beta$ = 0.841, significant at *p*-value (0.001).

The mediation role of corporate governance on CSR: As a mediator variable, corporate governance positively and significantly affects CSR. Assuming all things are constant, corporate governance with that of CSR ultimately enhances when the corporate board, ownership, audit, and transparency increase. When the engagement of better corporate governance on CSR increases by one unit, CSR's practice also improves by the estimated factor of $\beta$ = 0.776, significant at *p*-value (0.001). Generally, the TIRET corporate enterprise is highly affected by firm performance and the mediator variable corporate governance. The role of corporate governance influences firms' performance. The general model fit indices are shown in the following path diagram model in Figure 2.

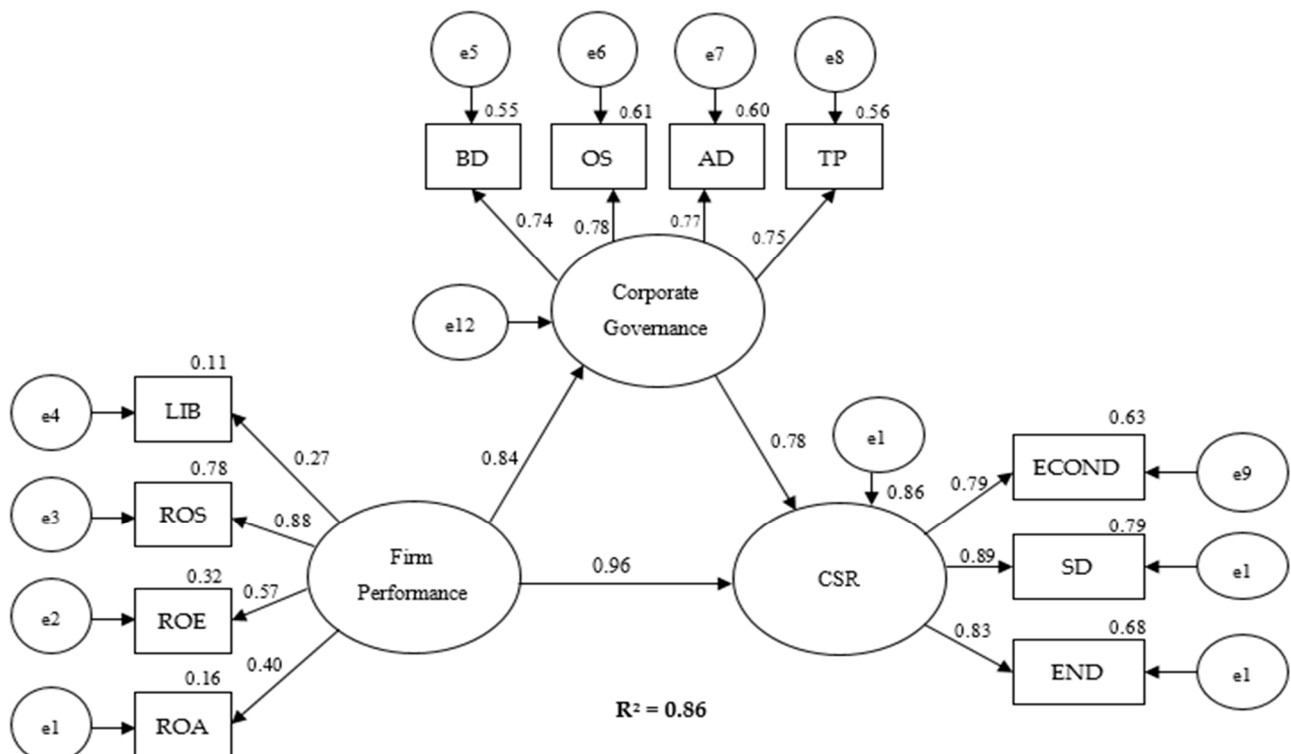

**Figure 2.** Parameter estimates of the structured model on CSR.

### 4.5.3. Hypothesis Test Analysis

The entire predicted hypothesis was derived from the literature and empirical studies. As the hypothesis test results show, the dependent variable firm performance positively and significantly affects CSR (standardized estimate β = 0.956 ***). Similarly, firm performance is affected by corporate governance, and corporate governance influences firms' performance positively and significantly (standardized estimate β = 0.841 ***). The mediator variable corporate governance directly and substantially affects CSR's outcome variable (standardized estimate β = 0.776 ***). All three variable measurement indicator variables are positively and significantly associated with each other. Hence, as shown in Table 7, the entire derived hypothesis was accepted.

**Table 7.** Summary of hypothesis test results.

| The Derived Hypothesis Direction & Structural Path | | | Standardized Estimate (β) Path Coefficient | S.E. | *t*-Value | *p*-Value | Decision |
|---|---|---|---|---|---|---|---|
| H₁—Firm Performance | -> | CSR | 0.956 | 0.289 | 5.181 | *** | Supported |
| H₂—Firm Performance | -> | Corporate Governance | 0.841 | 0.215 | 6.837 | *** | Supported |
| H₃—Corporate Governance | -> | CSR | 0.776 | 0.205 | 7.243 | *** | Supported |
| Explained Variance for the outcome variable CSR (R2) = 0.86 (86%) | | | | | | | |

Note: Firm Performance, Corporate Governance, CSR. *** Correlation is significant at the 0.001 *p*-value.

### 4.5.4. The Effects of Performance Indicators Ratio Analysis

As a collective investment, the previous twelve years of trend analysis growth paths show that the company's aggregate growth has been evaluated and projected based on total sales, total assets, total liabilities, and total capital for the period 2008–2020 [78]. The company evaluates its growth path in line with its strategic plan. The company's aggregate growth trends indicate that the total sales, assets, liabilities, and equity of the corporate company have increased. The result is a tremendous change, especially after the period 2015. In contrast, the company's total liability has been growing. As a result, the corporate enterprises' total assets are heavily dependent on debt financing. This implies that the firm may not have a more favorable environment to focus on CSR activities to improve firm performance due to the consequences of debt financing experience. It also indicates that if the company continues to do so, it is impossible to talk about CSR beyond modeling practices. Furthermore, this study used the corporate enterprises' growth path report to evaluate the impact of firms' performance on CSR and corporate governance. Accounting-based measures are primarily used for firm performance based on an enterprise's historical perspective and are limited to financial information from the past. As a result, the study used the accounting-based measures to analyze the performance indicators ratio estimation of ROA, ROE, ROS, and liability on capital structure (debt ratio), as shown in Table 8.

**Table 8.** Performance indicators ratio analysis and estimation.

| Indicators | Abbreviations | Financial Ratio Estimation |
|---|---|---|
| ROA | ROA | Net sales (Earnings before Interest & Taxes) total assets |
| ROE | ROE | Net income/Average equity ratio |
| ROS | ROS | ((sales revenue) − (operating expense))/Sales revenue |
| Liability (debt ratio) | LIB (DR) | Total liabilities/Total asset or 1−Equity ratio |

Source: authors' own computation.

The standardized correlation matrix results show a positive and significant relationship between the predictor firms' performance indicators with CSR and corporate governance. The matrix analysis provides the association, significant and estimated financial ratio analysis values, as shown in Table 9.

**Table 9.** Matrix of correlation coefficients of latent constructs and ratio analysis.

| Indicators | CSR | Firm. Per | Cog Gov. | ROA | ROE | ROS | LIB (DR) |
|---|---|---|---|---|---|---|---|
| CSR | 1 | | | | | | |
| Firm. Per | 0.591 *** | 1 | | | | | |
| Cog. Gov. | 0.469 *** | 0.660 *** | 1 | | | | |
| ROA | 0.312 *** | 0.731 *** | 0.305 *** | 1 | | | |
| ROE | 0.377 *** | 0.812 *** | 0.480 *** | 0.573 *** | 1 | | |
| ROS | 0.569 *** | 0.815 *** | 0.685 *** | 0.353 *** | 0.509 *** | 1 | |
| LIB (DR) | 0.588 *** | 0.662 *** | 0.594 *** | 0.137 *** | 0.310 *** | 0.711 *** | 1 |

Note: Financial ratio analysis for return on asset (ROA = 0.27), return on equity (ROE = 0.61), return on sales (ROS = 0.38), and liability on capital structure or debt ratio (DR = 0.39). The *** shows as all variables are strongly correlated.

As shown by the correlation matrix of panel data regression, the firms' performance indicators (ROA, ROE, ROS, and debt ratio) positively and significantly affect CSR and corporate governance. Moreover, the structural equation model analysis results confirm that ROA ($\beta$ = 0.40 ***), ROE ($\beta$ = 0.57 ***), ROS ($\beta$ = 0.88 ***), and liability on the capital structure or debt ratio ($\beta$ = 0.27 ***) affect firms' performance. Because of this, the path modeling analysis result also indicates that firm performance has a significant impact on CSR practice, with an estimated coefficient of $\beta$ = 0.961 ***.

With this in mind, the financial ratio analysis was computed to estimate each indicator ratio's levels and impact concerning firm performance, as shown in Table 9. As the financial ratio analysis, the estimated ratios of ROA imply that the company uses its assets



to generate profit or income. In this case, the calculated ratio of 0.27 shows that for every USD 1 in asset, the companies generate 27 cents in profit or revenue.

The estimated ratio of ROE shows the performance of the company. It indicates what the company is generating from its investment. From this perspective, the calculated ratio of 0.61 confirms that the company made USD 0.61 in profit for every USD 1 invested. Similarly, the estimated ratio of ROS tells the firms' efficiency. The ratio informs the company how much of each dollar of sales revenue remains after the company has paid the operating costs associated with generating that revenue. With these assumptions, the calculated ratio of 0.38 ROS shows that for every USD 1 in sales revenue, USD 0.38 remain after operating expenses.

Lastly, the estimated ratio of liability on capital structure (debt ratio) measures the proportion of the company's resources that rely on debt financing. The higher the ratio, the greater the risk associated with the firm's operation. A lower debt ratio indicates conservative funding with the potential to borrow in the future at no significant risk. In this regard, the calculated debt ratio is 0.39. It shows that the majority of the company's assets are financed by equity. The TIRET corporate company has a lower debt to equity ratio and higher equity to debt ratio. This illustrates that the majority of the company assets are supported by equity, because the debt to equity ratio is 0.39 and the equity to debt ratio is 0.61. If the debt to income ratio exceeds 0.50, the company faces operational and financial crises.

Generally, the study's findings assure us that corporate social responsibility in Ethiopia, the Amhara region, and in TIRET corporate firms is in an early stage. For this reason, the corporate firms have no conducive ground for CSR practice despite being engaged in social, environmental, and economic responsibilities on a compliance basis as state-owned enterprises. This conservative-based corporate social responsibility coupled with the absence of an organized CSR plan shows superficial understanding.

## 5. Conclusions

Companies have responsibilities towards society in the context of their business location and activities. As a result, corporate social responsibility (CSR) integrates social, economic, and environmental effects in their operations and the interaction with their anticipated stakeholders. Firms engage in CSR because they believe it will provide them with a competitive advantage. Hence, resource-based perspectives help to understand why firms to engage or do not engage in CSR activities and disclosure. From a resource aspect, CSR can be considered as generating both internal and external benefits. Firms that practice good social responsibility improve their external stakeholder connections as well as their employees' motivation, morale, dedication, and loyalty. Investments in socially responsible activities may provide internal use by assisting companies in developing new resources and capabilities. In this regard, it is difficult to think about CSR practices that incorporate business companies without jointly considering the impacts of firm performance and corporate governance. Hence, this study tried to identify and address the three determined research questions based on the stated objectives: (1) What is the influence of firm performance on corporate social responsibility practice? (2) How does the mediation role of corporate governance influence firms' performance and CSR practice? (3) Is there a relationship between the impacts of firm performance, CSR, and corporate governance?

According to the findings of this study, we conclude the theoretical implications as follows: CSR is a debatable issue, particularly in developing countries such as Ethiopia, including the study area, i.e., the Amhara region. As a result, firms' performance, corporate governance, and CSR are two sides of one coin. The practice of CSR is determined depending on firms' performance and the role of corporate governance. The research findings indicate that the outcome variable CSR is affected by firm performance, firm performance is influenced by corporate governance, and CSR is affected by the role of corporate governance with estimated factors of $\beta = 0.956$ ***, $\beta = 0.841$ ***, and $\beta = 0.776$ ***, significant at *p*-value $< 0.05$, respectively. Similarly, the indicator variables ROS, ROE, ROA, and liability

on capital structure strongly and significantly influence the company's performance. The indicator variables board, ownership, audit, and transparency positively and substantially affect corporate governance.

From the managerial point of view, the corporate firms' performance and governance have no more conducive ground to focus on CSR due to the impacts of debt on capital structure and the lack of return on exited assets related to the performance and governance gaps. Most of the corporate assets rely on fixed asset expansion and purchasing obsolete fixed assets due to an inefficient management system. Most of the available fixed assets have no direct linkage with the ultimate goals of the corporation. The other inevitable problems are foreign currency and power supply problems. The supply chain problems with stockholders are other bottlenecks. This research recommends that the company should identify unproductive enterprises and outsource non-core services. The corporation should minimize the capital expense on fixed asset expansion instead of increasing creation of other profitable assets. The adoption of CSR values results from some pressures or regulations to bring impact on development. To solve foreign currency problems, the enterprises should focus on export-based production. The government should give a special intervention to supply power energy and the corporation itself should provide other options. This study proposes that TIRET corporate enterprises should restructure, merge, rebrand, and reconsider existing business models.

*Limitations and Future Direction*

The limitations of this study open up new research areas. The sample was taken from TIRET corporate business enterprises in Ethiopia's Amhara region, which is one of the study's limitations. The study could be expanded to include other state-owned and non-state-owned firms at the regional and national levels for future research. In addition, the study's limitations can be seen in the context of CSR practices, the area of variable measurement, and the research objective scope.

This study proposes some future directions that take into account the potential synergistic effects of firm performance and CSR practices. The TIRET Corporation is currently a state-owned, monopoly, and non-competitive business sector. Due to its undistinguished ownership structure and monopoly practices, it does not play its expected role. To make use of social responsibility for better development, it needs increasing state intervention. At the corporate company level, all the 21 enterprises should be technology-based firms using advanced technology such as bitcoins to manage, evaluate, and sustain efficient operations rather than using a traditional manual system.

However, this research is significant. It adds to the CSR literature in the context of emerging economies. Firms, policymakers, and practitioners may take steps to improve CSR practices. Moreover, the contribution of this study on corporate social responsibility will fill gaps in the use of corporate governance as a mediator role between the impacts of firm performance and CSR practices, which other studies have not explored broadly.

**Author Contributions:** Conceptualization, M.Y., H.S. and G.A.T.; methodology, software, validation, and formal analysis, investigation, resources, data curation, writing—original draft preparation, G.A.T.; writing—review, editing and supervision, M.Y. and H.S. All authors have read and agreed to the published version of the manuscript.

**Funding:** This research received no external funding sources.

**Institutional Review Board Statement:** Not applicable.

**Informed Consent Statement:** Not applicable.

**Data Availability Statement:** The data that support the findings of this study are available from the corresponding author upon reasonable request.

**Acknowledgments:** We appreciate all supports from School of Management, Wuhan University of Technology. The authors would like to thank all those who participated in the survey for their valuable inputs.

**Conflicts of Interest:** The authors declare no conflict of interest.

**Originality Statement:** We declare that this is our own original work based on authors' knowledge, containing no materials previously published or written by another person. Furthermore, the authors gathered data, materials, and analyzed them using a unique understanding of an integrated research model in the context of Ethiopia, specifically the TIRET corporate business state-owned enterprises in the Amhara region.

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
