# Peer review of "Impacts of Firm Performance on Corporate Social Responsibility Practices: The Mediation Role of Corporate Governance in Ethiopia Corporate Business"

_sustainability, doi:10.3390/su13179717_

Round 1

Reviewer 1 Report

The study is interesting, well-structured and up to date for the economic environment in Ethiopia.

The introduction provides information regarding the importance of the topic and the state of arts literature in the fields as well as importance of the approach for the specialist and for decision markers. The research questions are well formulated and provide a clear imagine  of the author's approach. We suggest to accentuate the elements of originality of the work paper.

The paper is well structured , providing information on the evolution of the CSR literature based on a well selected bibliography.

The methodology is clear and adequate to the author's approach. We consider necessary the inclusion in fig. 1, of the symbol of the variables that will be used later in order to facilitate the understanding of the text.

The results of models are well presented and clear explained.

The conclusions are logical. We suggest to connect on a clear manner the conclusions with the research questions.

Author Response

Thank you for giving us the opportunity to revise our manuscript entitled "Impacts of Firms' Performance on Corporate Social Responsibility Practices: The Mediation Role of Corporate Governance in Ethiopia Corporate Business" for the Journal of Sustainability. We appreciate your time, effort, critical and powerful criticisms on our manuscript for the insightful comments and significant improvements to our paper. Furthermore, all of the questions, comments, and suggestions provided are very valuable in improving the quality of our article.

Reviewer 2 Report

- interesting topic and field of research
- I would suggest to at least also discuss the possible vice-versa influence of CSR on companies performance, which has been studied multiple times - is it a coincidence or is it correlating? Why did the authors approach it in another way? Why is a company which is more successful more likely to apply CSR practices? This is discussed in the introduction, but unclear in the abstract 
-difference between corporation and enterprise is unclear
- please apply proof reading - for example line 22 or 125 or 315 or 651 or 684
- CEO in line 63 is unclear
- I would suggest to also incorporate the concept of dynamic capabilities in the introduction (first part)
- Is the model / approach also adaptable for other countries in the world? Which limitations can be drawn? 
- Why did the authors chose Ethiopia as the country for research? This is not clearly stated. 
- Please elaborate more clearly how the hypothesis and the research questions are interconnected.
- Could there be a way to reduce the number of research questions to one for the sake of simplicity? 
- Please add literature in the first part of the introduction for better reference
- I would suggest that the authors add a c.p. remark to this study / to the model
- Please review if the arrows in figure 1 are really pointing into the right direction (according to your research approach, yes, but according to the recent research) - especially I would see ROA etc impacting firm performance and not the other way round (firm performance being an agregator variable) - please discuss this. Consider the renaming of the figure if applicable. 
- Please strengthen the methodological discussion with more literature on the methods used and additional cross references to similar approaches in other studies
- I strongly recommend to explain and elaborate on the TIRET sector in more detail. Why has it been chosen? What influence does it have etc? 
- in the current study design it is not obvious that the questionnaire has been distributed among the corporation observed. It is suggested to include the questionnaire in the appendix of the study.
- the results seem legit and also offer interesting insights 
- observing the research findings, is it questionable if the findings are too obvious? Was the study critical and extensive enough? It is suggested to include these points into the limitations of the study
- it is recommended to discuss in the conclusion if it is not a logical consequence that profitable companies can afford CSR activities while firms struggling for existence are not engaging into CSR actions. 
- Can this study be also expanded to non-state owned companies? Please provide future research perspectives in the conclusion. 
- it is strongly suggested to add theoretical implication as well as managerial perspectives into the conclusions. 
- furthermore, the limitations of the study should be extended in the conclusions. 

Please evaluate this literature for inclusion (suggestion):
https://doi.org/10.1177%2F0312896218771438
https://doi.org/10.3390/su12093514
http://dx.doi.org/10.3390/jrfm14020049
https://www.sciencedirect.com/science/article/pii/S2092521219300057
http://dx.doi.org/10.3390/jrfm14020087
https://doi.org/10.1002/kpm.1616

Author Response

Thank you for giving us the opportunity to resubmit with a major revision of the manuscript entitled "Impacts of Firms' Performance on Corporate Social Responsibility Practices: The Mediation Role of Corporate Governance in Ethiopia Corporate Business" for the Journal of Sustainability. We appreciate your time, effort, critical and powerful criticisms on our manuscript for the insightful comments and significant improvements to our paper. Furthermore, all of the questions, comments, and suggestions provided are very valuable in improving the quality of our article.

Reviewer 3 Report

The presented article for review discusses a very important problem from the scientific point of view. The subject matter of the article is and has been researched many times by scientists all over the world. So it cannot be considered a fresh and innovative idea. Nevertheless, the article deserves attention.

This study attempted to identify and answer the following three core research questions based on specific goals. (1) What driving model influences the company's performance in terms of corporate social responsibility practice? (2) How does the mediating role of corporate governance affect corporate performance and CSR practices? (3) Is there a link between the impact of company performance, CSR, and governance?

My comments to the article:

1) The work contains one research hypothesis: H1: CSR has a significant and positive impact on firms' performance - which is correctly formulated, but I have doubts whether there should be more hypotheses concerning the aim or rather the aims of the article?

I do not like this formulation of the hypotheses:
H1: Firms' performance has positive and substantial impacts on CSR practice. 258
H1: a: Return on asset positively and significantly affect firms' performance. 259
H1: b: Return on sales positively and significantly influences firms' performance. 260
H1: c: Return on equity positively and significantly affects firms' performance. 261
H1: d: Liability or debt ratio negatively and significant influences firms' performance

Can it be improved and can it be generalized? Honestly, I got lost in the numbers of hypotheses and their numbers.

for example another ...

H1: Corporate governance positively and significantly influence firms' performance.

and more

H1: Corporate governance positively and significantly mediates firm performance and CSR.
H1: a: Corporate governance has a direct effect on firm performance and CSR.
H1: b: Corporate governance has an indirect effect on firm performance and CSR.

Is it possible to sort it out logically? It is better to make one or a maximum of 3 hypotheses and prove or deny them.

2) I have reservations about the statement: Conceptual Framework of the Study - point 2.4. At the moment, this research cannot be called frameworks. Can you call it any other way?

3) Chart 1 needs to be supplemented. He suggests that the performance companies are only ROA, ROS, ROE, and obligations. Or it suggests that these values ​​are the most important thing that is not true. In addition, the elements of CSR extend significantly beyond the 3 areas listed in the chart.

4) pattern editing is not correct. My question is how these formulas are formulated. What was their process of creating what are the assumptions and limitations? Any regression model must be constructed in some way dictated by econometrics. Please explain to me what about the residuals, what about the autocorrelation. What tests were used?

5) Please explain the variable selection.

6) Were model force tests performed?

please reply.

Author Response

(The authors gave the same response as above.)

Round 2

Reviewer 2 Report

Thank you for your revision. Though some of the remarks could have been incorporated a bit more clearly, I think, the paper is now ready for being published. It is an interesting and good addition to the scientific discussion.

Reviewer 3 Report

I accept all corrections.